**Data Availability Statement:** All study data and analysis files are available from the Open Science Framework, DOI: 10.17605/OSF.IO/BFXNV.

**Funding:** This research was partially funded by a grant (R01 HD080292-02) from the Eunice

# Translating episodic future thinking manipulations for clinical use: Development of a clinical control

**Jillian M. Rung**◯¤*, **Leonard H. Epstein**

Department of Pediatrics, University at Buffalo School of Medicine and Biomedical Sciences, Buffalo, NY, United States of America

¤ Current address: Department of Epidemiology, University of Florida, Gainesville, FL, United States of America
* rung.jillian@gmail.com, jrung@ufl.edu

## Abstract

Many studies support that Episodic Future Thinking (EFT) reduces maladaptive health behaviors and how much individuals devalue the future (steepness of delay discounting). In order to understand the clinical utility of EFT, a control procedure that equates groups in non-specific treatment factors (e.g., expectancy of change, perceived connection of content to health behavior) is needed. The present research evaluated the effects of EFT relative to a novel control (health information thinking; HIT), which was designed to be structurally similar to EFT while incorporating elements from existing clinical controls. In a sample of Amazon Mechanical Turk workers ($N = 254$), we found that EFT reduced discounting relative to the HIT procedure and a standard EFT control. There were some affective differences across groups and differences in adherence to the intervention content, but these were unrelated to discounting. Delay discounting was not equivalent across the control groups, but this may not be a necessary condition to fulfill for a clinical control. Future research will need to identify whether the HIT procedure serves as a good control for other dependent variables when studying EFT and develop controls analogous to usual care or a "wait-list" in clinical contexts.

## Introduction

Many public health issues can be conceptualized as arising from chronic preference for smaller-immediate outcomes in lieu of larger, delayed outcomes. For instance, repeated consumption of high-fat, high-sugar foods relative to more nutritious, less energy-dense foods leads to weight gain and obesity; and repeated use of addictive drugs leads to dependence and lower quality of life. This pervasive preference for immediacy is referred to as steep delay discounting [1]: the value of these otherwise greater alternatives (e.g., longer life span, better health, financial, and social well-being) is reduced because of their delay [2].

Because steep discounting is prevalent in many behavioral health concerns [3], researchers have begun investigating ways to reduce delay discounting. One method that has proven

Kennedy Shriver National Institute of Child Health and Human Development (NICHD), of the National Institutes of Health (NIH), awarded to Leonard H. Epstein. Jillian M. Rung's time in preparing this report was partially supported by a fellowship at the UF Substance Abuse Training Center in Public Health (T32DA035167) from the National Institute on Drug Abuse (NIDA) of the NIH. The funders had no role in study design, data collection and analysis, decision to publish, or preparation of the manuscript.

**Competing interests:** The authors have declared that no competing interests exist.

successful is episodic future thinking (EFT), which involves the imagination, or pre-experiencing of one's future [4, 5]. A variety of laboratory studies support that EFT reduces discounting; EFT also improves the maladaptive behaviors linked to long-term poorer outcomes (e.g., smoking, caloric intake [6, 7]). Because of the success of EFT thus far, work is underway to develop and test EFT as a clinical intervention in the field [8, 9].

In laboratory settings, the effects of EFT on delay discounting are often evaluated by prompting individuals to think about future events via the use of cues while they are completing a delay discounting assessment [6, 10, 11]. For example, participants are asked to identify several possible, positive future events and to think about them in an episodic manner. To standardize the amount of time into the future individuals are prospecting, participants are given future dates or delays as anchors, and names and descriptions of participants' identified events are recorded. Then, participants are presented their cues (and/or event descriptions) in auditory (e.g., recording of the participant reading their cue name or description) or visual format (e.g., a textual prompt or cue card) while completing a delay discounting task. Participants are instructed to imagine the event prompted in the cue/description while engaging in the task. This type of procedure has been used not only for evaluating the effects of EFT on delay discounting, but also on demand for drugs and food [11, 12], as well as real consumption of the aforementioned [6, 7]. In initial clinical application for weight reduction [9], EFT has been conducted similarly with prompts to imagine events delivered close to meal times via text message or e-mail.

As EFT investigations shift to a more clinical focus, it is important to develop and test control procedures that are feasible in clinical settings. To date, the most widely used control for EFT is episodic recent thinking (ERT [7, 10, 11, 13]), which was designed to evoke episodic imagination and hold constant the personal relevance and vividness of imagined events [14]. ERT is typically conducted in a parallel manner to EFT. Importantly, keeping aspects such as the episodic nature of content constant across groups allows for strong conclusions that EFT effects are a result of future thinking. Other less commonly used control procedures delineate the importance of episodic imagination while controlling for future orientation of thought, such as simply estimating what future money could be spent on (e.g., see Benoit, Gilbert, and Burgess [15]).

ERT is a well-designed theoretical control, but its use in clinical settings may be problematic. First, and as previously described, EFT and ERT are typically employed by having individuals complete a cue/event description generation procedure, from which prompts to engage in EFT/ERT are derived. If researchers wish to maintain the temporal proximity of ERT cues (e.g., for an event that happened one day ago) throughout the course of treatment, participants would need to regenerate their events and cues on a daily or otherwise frequent basis. Constant cue creation in an intervention that would likely need to be implemented for at least several weeks to identify meaningful changes in behavior (see Sze, Daniel, Kilanowski, Collins, and Epstein [9] for an example), would be burdensome both for the participant and the experimenter (i.e., daily regeneration of cues [plus review of cues by staff for meeting criteria], active use of cues throughout the day, in addition to other tasks/intervention content as relevant to the trial). Second, the frequency of cue creation would ideally need to be matched across control and EFT groups. While there would continuously be new and easily accessible content for individuals completing ERT due to the simple passage of time, imagining the same number of distinct future events at a high frequency could be challenging for those assigned to EFT. Third, while ERT cues could be "recycled" and used despite becoming older, retrospection relies on the same neural circuits as prospection [16]; allowing more distant retrospection, then, could inadvertently elicit future thinking or the benefits thereof [17]. Finally, in clinical studies a strong control group holds constant things such as contact with the participants and

instills a minimum expectancy of change/improvement—in other words, control procedures should match treatments in their arrangement of non-specific treatment factors (see Borkovec [18]). It seems unlikely that ERT would suffice to impart these non-specific factors, as participants do not anticipate ERT to lead to improvements in health behaviors (or reductions in discounting), while EFT does [19, 20].

To properly evaluate the effects of EFT on health behaviors in clinical applications, the development of controls better suited to that context are needed. The goal of the present research was to develop and evaluate a novel EFT control procedure that is more feasible in, and better meets the needs of clinical research. The novel control tested herein is health information thinking (HIT), which involves providing participants health-related information. This procedure both standardizes the content participants will think about and mimics psychoeducational controls in the field (e.g., nutrition-related info as a control for assessing EFT effects on weight loss [9]), which is better suited to equate the procedures in expectancy of behavior change in clinical contexts by increasing the connection between intervention content and health. Importantly, this research serves as a translational proof-of-concept for subsequent testing and development in clinical samples.

## Materials and methods

### Participants

Two-hundred fifty-four participants were recruited online, on Amazon Mechanical Turk ® (AMT). AMT is a web-based crowdsourcing platform in which individuals can complete tasks for pay. AMT users were eligible to participate if they were at least 18 years of age, had a 90% or greater approval rating (i.e., 90% of their previously completed AMT tasks have been approved, which is synonymous with paid), had at least 100 AMT tasks approved, and resided in the United States.

Participants learned of the study through a study-specific listing on AMT, which included a brief description of the purpose of the study, the compensation ($5.00) and bonus amounts ($2.00), and the criteria for task approval and receipt of bonus. Participants' tasks/survey completions were approved on AMT contingent on the following criteria: the participant reasonably attempted to follow the survey instructions (i.e., responses to open-ended questions at least minimally addressed the respective prompt), provided a valid survey completion code (retrieved at the end of the study survey), and completed the survey from within the United States. The latter was determined based on participants' IP addresses. Bonuses were awarded to those who met the aforementioned approval criteria and passed at least one attention check (details provided below; all participants in the analytic sample were awarded the base and bonus compensation).

All participants were provided a consent document outlining study procedures and subsequently provided informed consent to participate by selecting a survey response option that said "I agree to the terms above," which was shown immediately below the consent document. No electronic signatures were required because they constitute identifying information; collecting such information is in violation of terms of use of Amazon Mechanical Turk. This research was approved by the University at Buffalo Institutional Review Board and appears in the protocol named, "Equivalencies of Thoughts," protocol #00002838.

### Measures

All measures were embedded into an online survey hosted by Qualtrics. The survey began with either Episodic Future Thinking (EFT), Episodic Recent Thinking (ERT), or the novel control task, Health Information Thinking (HIT). The EFT and ERT tasks were based on

those in other published papers from our laboratory [13, 17, 21]. In both the EFT and ERT tasks, individuals were asked to vividly imagine 6 different events. Those assigned to EFT were asked to identify positive future events that were going to, or could reasonably occur at different delays (2 weeks, 1 month, 3 months, 6 months, 1 year, and 2 years from now); and those assigned to ERT were asked to imagine positive, recent events that occurred in the last week (12 hours ago, 1 day, 2 days, 3 days, 4 days, and 5 days ago). Participants identified their events by completing the following sentence (writing the ending in an open text-box): "In 1 day from now, I am/About 12 hours ago, I was. . ."

After identifying an event for a given time period, individuals rated their events on several characteristics and then elaborated on them in episodic detail. Specifically, individuals rated their events for (1) how much the participant likes the event, (2) how important (3) and exciting the event is, (4) how well/easily the associated details of the event can be imagined, and (5) how vividly the participant can imagine the event. Ratings were on a scale of 1 (not at all) to 5 (very much). Then, participants were prompted to imagine each event in detail and to describe in several sentences who they were with, what they were doing, where they were, and how they were feeling.

Those completing the HIT procedure were asked to read six paragraphs pertaining to different health topics, and then to describe in one sentence a specific piece of information they learned about from each presented topic. Participants read and wrote about these different health topics one at a time. The latter portion (information identification) was designed to mimic the event identification phase in the EFT/ERT procedures. Using the same scale as those assigned to EFT/ERT, participants provided ratings about the specific information they learned after identification: (1) how much they liked learning the information, (2) how important learning it was, (3) how exciting it was to learn the information, and (4) how useful the information is. After providing ratings, individuals were asked to describe learning about the specific piece of information they noted, and to address (1) how the information fits into their existing knowledge, (2) what the information made the participant think about, (3) who (and what) the information may be useful for, and (4) how the information made them feel.

The six paragraphs participants read consisted of information pertaining to alcohol's effects on sleep, depression, prediabetes, electronic cigarette use, nutrition labeling, and physical activity. These different topics and their contents were chosen on the basis of two primary considerations. First, the topics could reasonably appear in a psychoeducational control (depending on the clinical population/behavior of interest). Second, the topics contained information that was either relatively new (e.g., changes in guidelines for nutrition labeling and recommendations for physical activity) or likely to be lesser-known (sex differences in the expression of major depressive disorder, rebound alertness of alcohol interfering with sleep). The six paragraphs of information that were presented to participants are provided in the S1 File.

Participant's descriptions of EFT, ERT, and health information were later displayed in the discounting assessment. The single sentence event/information identifications are hereafter referred to as *tags* and these sentences, concatenated with the longer elaborations, are referred to as *cues* for the purposes of the discounting task below.

**Delay discounting task.** Participants completed an adjusting immediate amount delay discounting task (e.g., [22]), which consisted of a series of choices between smaller-immediate and larger-delayed monetary outcomes. For the first trial in a block, participants were asked to indicate their preference between $50 now or $100 after a delay. With each selection, the smaller, immediate amount of money was adjusted up (if choosing the larger) or down (if choosing the smaller) on the following trial. Adjustments started at half the magnitude of the smaller immediate amount of money ($25) and continued to decrease by half each subsequent trial, for a total of 6 trials per larger outcome delay. The delays to the larger amount of money,

presented in ascending order across consecutive blocks, were the same as the timeframes of EFT cues: 2 weeks, 1 month, 3 months, 6 months, 1 year, and 2 years.

Two attention checks were embedded in the discounting task, which were in the form of additional trials in two separate delay blocks. The first check asked individuals to choose between $0 now and $100 now, and appeared in the 1-month block as its second-to-last trial; and the second check asked individuals to choose between $500 now and $100 in 1 year, which appeared in the 1-year block as its third-to-last trial. Because very few participants in the analytic sample responded incorrectly to the attention checks in the discounting task (only 11 [6%] participants failed one check, and none failed both), no participants' data were excluded from analyses of discounting data on the basis of these questions.

Prior to beginning the discounting task, participants were instructed that the monetary outcomes presented in the task were hypothetical, and to simply pick the outcome they would prefer. At the start of each new block of trials, participants were oriented to the delay of the larger delayed monetary outcome and asked to think about their corresponding EFT/ERT event or recently learned health information. Participants were shown their EFT cue that matched the delay of the larger-later outcome, the ERT cue that matched the ordinal temporal distance of the larger-later outcome (e.g., the most recent cue was shown for the first delay, the second most recent for the second delay), or their HIT cue that followed the order of that described in the HIT procedures above. Thereafter, on each trial in the discounting task, participants were reminded to think of their events/information by displaying the corresponding tag above the choice options with a prompt that said, "Keeping [EFT/ERT/HIT tag] in mind: Which of the following options would you prefer?" At the end of each block, participants were asked how vividly and how frequently they imagined their events while completing that particular block of trials.

**Positive and negative affective schedule (PANAS).**   To assess if the novel control procedure induced a greater degree of negative or positive affect, the PANAS was used [23]. The PANAS is a validated, 20-item scale that asks individuals to rate how much they have felt 10 different types of positive and negative emotions or states over a given time span. For the purposes of the present research, individuals were asked to indicate how much they were feeling the different emotions/states right now. Items are rated on a 5-point scale ranging from Very Slightly or Not at All (1) to Extremely (5). Typically, the sum of the scores on the positive items and the sum of the scores on the negative items form two subscale scores for positive and negative affect, respectively. Because some participants skipped items, we used the average item score. Participants who skipped more than 2 items each on the positive and negative subscales were excluded from relevant analyses including the PANAS ($n$ = 4; 2 participants each from the EFT and ERT groups) but retained in all others. I.e., if a participant skipped three positive subscale items, they were excluded from analyses involving positive, but not negative affect.

**Demographics.**   Participants were asked to provide basic demographic data such as sex, race/ethnicity, yearly income, and education (highest level achieved and years completed). Participants also indicated their subjective socioeconomic status [24], height and weight (to calculate BMI), their prior experiences completing EFT/ERT procedures or a delay discounting task, and several questions pertaining to the circumstances of working on the survey (multitasking, where they were working on the survey). Participants were able to elect not to respond to the aforementioned questions; 29 participants chose not to disclose their income (7–13 participants per group), 34 chose not to disclose either their height, weight, or both (11–12 per group); non-response was rare across other non-categorical items (range = 1–2 missing values). Of the total sample, 11 participants did not disclose whether they were multitasking (3–4 per group); 2 participants did not disclose their prior experience with EFT/ERT tasks (2

participants in 1 group); and 2 did not disclose their prior experience with a discounting task (1 participant each in the EFT/ERT groups).

**Procedures.** Within AMT, interested individuals first accepted the HIT, then after reviewing and providing informed consent, began either EFT, ERT, or the HIT procedure (randomly assigned). After, individuals completed the discounting assessment with their previously generated tags/cues, followed by the PANAS and demographics measures. Participants then received their survey completion code and submitted it on AMT to receive compensation for completing the survey.

**Sample size determination.** The sample size was determined based on prior EFT research conducted on AMT. The effect size of EFT (relative to ERT) on discounting in AMT samples ranges from approximately $d = 0.35$ to 1.46 (effect sizes calculated based on data in graphs and/or means and *SD*s [11, 12, 25]). Using an effect size of $d = 0.50$ and power of .80, approximately 64 participants per group are needed to detect a significant difference in discounting between EFT and ERT groups (analysis conducted in G*Power [26]). We anticipated participants would evidence a similar degree of discounting across ERT and HIT and thus did not base sample size estimates on any additional calculations.

**Data analysis.** Responses from open-ended questions in EFT/ERT/HIT were subjected to screening procedures prior to analysis. Specifically, participants' cues were reviewed for addressing the questions in the assigned condition. Participants' data were flagged for exclusion (and rejection on AMT) if their EFT/ERT/HIT cues were unrelated to the question/prompt (e.g., did not mention an event or information presented in the relevant HIT paragraph), consisted primarily of text copy/pasted from example cues and/or HIT paragraphs, or contained otherwise incomprehensible responses (see Rung and Madden [20]).

Prior to primary analyses of discounting data, differences across groups on demographic, thought-based (mean frequency of thoughts and vividness thereof), and affective measures were assessed. Variables that differed across groups, based on the results of Kruskall-Wallis and Fisher's exact tests (as appropriate), were included as covariates in analyses of discounting data. For the aforementioned analyses involving non-categorical variables, outliers were classified as observations exceeding 3x the IQR; when outliers were detected, they were removed and the analysis was re-run. Results of analyses post-removal of outliers are reported only in cases where it affected the outcome of a test (i.e., whether a comparison was significant or not). All statistical tests were conducted using *R* [27] and the type of tests chosen were based on the results of tests of normality (Shapiro-Wilks) and/or the data type as appropriate (e.g., for continuous vs. categorical variables).

Steepness of discounting was quantified as the proportion of area under the curve (AUC). AUC is calculated by summing the trapezoids formed by indifference points when plotted as a function of delay and dividing by the total possible graphical area (see [28]). The effects of EFT relative to ERT and HIT on AUC were then examined using beta regression (via the betareg package [29]). Beta regression is a type of generalized linear model appropriate for bounded, proportion data [29, 30]. Test statistics for main effects were obtained using the car package [31] and pairwise comparisons with the lsmeans package [32]. Pairwise comparisons were corrected for multiple comparisons using the False Discovery Rate method [33]. Effect sizes (Cohen's *d*) are based on all raw observations and were calculated using the effsize package [34].

Model diagnostics were conducted to identify large residuals (>3 *SD*s) and overly-influential values (Cook's *D*; using a cut-off of 4/*N*). When values meeting the threshold for either of the aforementioned were found, they were removed from the analysis and the model was re-run. In all cases, the nominal results were unchanged by the removal of problematic observations. The *p*-values for pairwise comparisons in the post-diagnostic models either remained at

the same level of significance (e.g., $p < .05$) or were significant at a more stringent threshold (e.g., $p < .01$) relative to models containing all observations. Thus, for the sake of brevity, pairwise comparisons from post-diagnostic models are not reported herein.

To ensure group differences (or lack thereof) were not due to other extraneous variables, secondary regression models including covariates were conducted. Several models were conducted either with individual, or small groups of covariates in addition to Group as a predictor (covariates based on outcomes of tests mentioned above). These secondary analyses were conducted across multiple models to balance concerns with over-fitting while attending to the potential relevance of covariates in the effects of EFT relative to the control procedures. As with the primary models above, the results of these secondary models were nominally the same post-diagnostics; results from initial but not post-diagnostic models are reported herein.

Finally, the similarity of AUCs across the HIT and ERT groups were characterized with an exploratory equivalence analysis using Bayesian estimation. A posterior distribution of mean differences in AUC was computed using Markov-Chain Monte Carlo sampling from participants' obtained AUC values (i.e., the mean differences in AUC were estimated by resampling the obtained data). A threshold of +/- 0.10 (the region of practical equivalence) served as the interval within which the most likely mean differences (called the credible interval) must lie to be considered equivalent. A threshold value of .10 is less than the difference between EFT and ERT groups (obtained *Mdn* difference = .13) and is the same as that used in other research [20]. Equivalence was tested using both a 95% and 90% credible interval; in other words, 90–95% of the most likely mean differences needed to fall within -.10 and + .10 for AUCs to be considered equivalent across the two groups. The equivalence testing was completed using the BEST package [35].

## Results

Of the 254 participants who completed the study, 15 (5.9%), 15 (5.9%), and 22 (8.7%) participants from the EFT, ERT, and HIT groups were excluded for not appropriately completing the assigned thought procedures. An additional 4 (1.6%) were excluded from the ERT group for completing the survey outside of the United States. Thus, the primary analytic *N* was 198. The number of participants assigned to each of the groups, as well as descriptive statistics of demographic variables, are in Table 1. Groups did not differ significantly on any of the demographic or individual difference characteristics assessed (all $ps \geq .18$). A relatively small percentage of total participants reported multitasking (8%), completing the HIT outside of their home (8%), and having prior experience completing EFT/ERT procedures (28%). A larger percentage had prior experience completing delay discounting tasks (58%; 55–62% of participants in each group). Because there was insufficient variability (e.g., multitasking) to meaningfully analyze the potential impact of contextual variables for survey completion, these data were not analyzed. Discounting task and episodic thinking manipulation experience were unrelated to degree of discounting within each of the groups (all $ps \geq .15$) and as such they are not further discussed or included in analyses herein.

The Kruskall-Wallis test revealed that the different thought-based tasks took different amounts of time to complete ($\chi^2[2] = 44.70$, $p < .001$), and in some cases yielded differences in the characteristics of thoughts produced. In particular, the HIT procedure took participants significantly longer to complete (*Mdn* = 47.08) than the ERT (*Mdn* = 25.98 mins; $W = 854$, $p < .001$) and EFT procedures (*Mdn* = 32.48 mins; $W = 979.5$, $p < .001$); there were no differences in durations between the EFT and ERT groups ($p = .21$). Also revealed through Kruskall-Wallis tests, the thought tasks generated different degrees of positive ($\chi^2[2] = 7.50$, $p = .023$) and negative affect ($\chi^2[2] = 6.06$, $p = .048$) across groups. Specifically, the HIT procedure

**Table 1. Demographic variables for participants in EFT, ERT, and HIT groups (N = 198).**

| Variable | EFT (n = 62) n (%) | EFT (n = 62) Median (Q1-Q3) | ERT (n = 71) n (%) | ERT (n = 71) Median (Q1-Q3) | HIT (n = 65) n (%) | HIT (n = 65) Median (Q1-Q3) | p |
|---|---|---|---|---|---|---|---|
| Age (years) | | 34 (29–43) | | 35 (28–45) | | 36 (28–47) | .83 |
| Sex | | | | | | | .18 |
| Male | 32 (51.6%) | | 29 (40.8%) | | 34 (52.3%) | | |
| Female | 28 (45.2%) | | 41 (57.7%) | | 31 (47.7%) | | |
| Other | 2 (3.2%) | | 0 (0) | | 0 (0) | | |
| N/A | 0 (0) | | 1 (1.4%) | | 0 (0) | | |
| Race/Ethnicity | | | | | | | .61 |
| American Indian/ Alaska Native | 1 (1.6%) | | 0 (0%) | | 0 (0%) | | |
| Asian | 2 (3.2%) | | 4 (5.6%) | | 3 (4.6%) | | |
| Black/African American | 7 (11.3%) | | 4 (5.6%) | | 11 (16.9%) | | |
| Hispanic or Latinx | 1 (1.6%) | | 3 (4.2%) | | 2 (3.1%) | | |
| Multiple | 4 (6.5%) | | 3 (4.2%) | | 2 (3.1%) | | |
| Native Hawaiian/ Pacific Islander | 0 (0.0%) | | 0 (0.0%) | | 0 (0.0%) | | |
| White | 47 (75.8%) | | 57 (80.3%) | | 47 (72.3%) | | |
| N/A | 0 (0.0%) | | 0 (0) | | 0 (0) | | |
| Education (years) | | 16 (14–16) | | 16 (13–16) | | 16 (13–16) | .99 |
| Income (units of $10k) | | 38 (21–60) | | 34 (21–56) | | 36 (25–48) | .93 |
| Subjective SES (Community) | | 5 (4–6) | | 5 (4–6) | | 5 (4–6) | .87 |
| Subjective SES (Country) | | 5 (4–6) | | 5 (3–6) | | 5 (4–6) | .95 |
| BMI | | 26 (23–30) | | 24 (22–30) | | 26 (22–30) | .50 |

Percentages do not always sum to 100% due to rounding. Age and years of education are rounded to the nearest year, income to the nearest $1,000 increment, and BMI the nearest whole value. Participants with BMI values below 15 and above 60 (n = 4; range 3.26–14.99) were excluded from descriptive and inferential statistics.

resulted in significantly lower positive affect (*Mdn* = 3.50) compared to EFT (*Mdn* = 3.95; *W* = 2505.5, *p* = .006) and nominally so compared to ERT (*Mdn* = 3.80; *W* = 2649, *p* = .07). There were no significant differences in positive affect between the EFT and ERT groups (*p* = .457). The HIT procedure (*Mdn* = 1.10) did not result in a different degree of negative affect relative to the EFT (*Mdn* = 1.10) or ERT (*Mdn* = 1.00) groups (*ps* ≥ .055) using all available data; although negative affect significantly differed between HIT (*Mdn* = 1.2) and ERT (*Mdn* = 1.1) after the removal of outliers (*n* = 14), *W* = 1537, *p* = .01. EFT resulted in slightly greater negative affect than ERT (*W* = 2633, *p* = .022).

The application and characteristics of the assigned thought content differed during the discounting assessment. There were significant differences in both how vivid ($\chi^2[2]$ = 7.15, *p* = .028) and frequent cue-associated thoughts occurred ($\chi^2[2]$ = 9.47, *p* = .009). Vividness was significantly lower in the HIT group (*Mdn* = 4.00) relative to EFT (*Mdn* = 4.50; *W* = 2545, *p* = .010), and nominally so relative to ERT (*Mdn* = 4.33; *W* = 2733.5, *p* = .061). Frequency of cue-associated thoughts was also lower in the HIT group (*Mdn* = 4.0) relative to those in the EFT group (*Mdn* = 4.33; *W* = 2588, *p* = .005) and ERT groups (*Mdn* = 4.33; *W* = 2876.50 *p* = .012). There were no significant differences in vividness (*p* = .442) or frequency (*p* = 0.70) of cue-associated thoughts between the EFT and ERT groups.

The discounting-reducing effect of EFT was replicated both in primary analyses confined to evaluating group-level differences, as well as in secondary analyses in which thought-based or–induced characteristics were controlled for (e.g., positive and negative effect, frequency of thoughts). Fig 1 shows the AUCs for participants assigned to each of the groups, along with the group medians and interquartile ranges. The parameter estimates from the primary

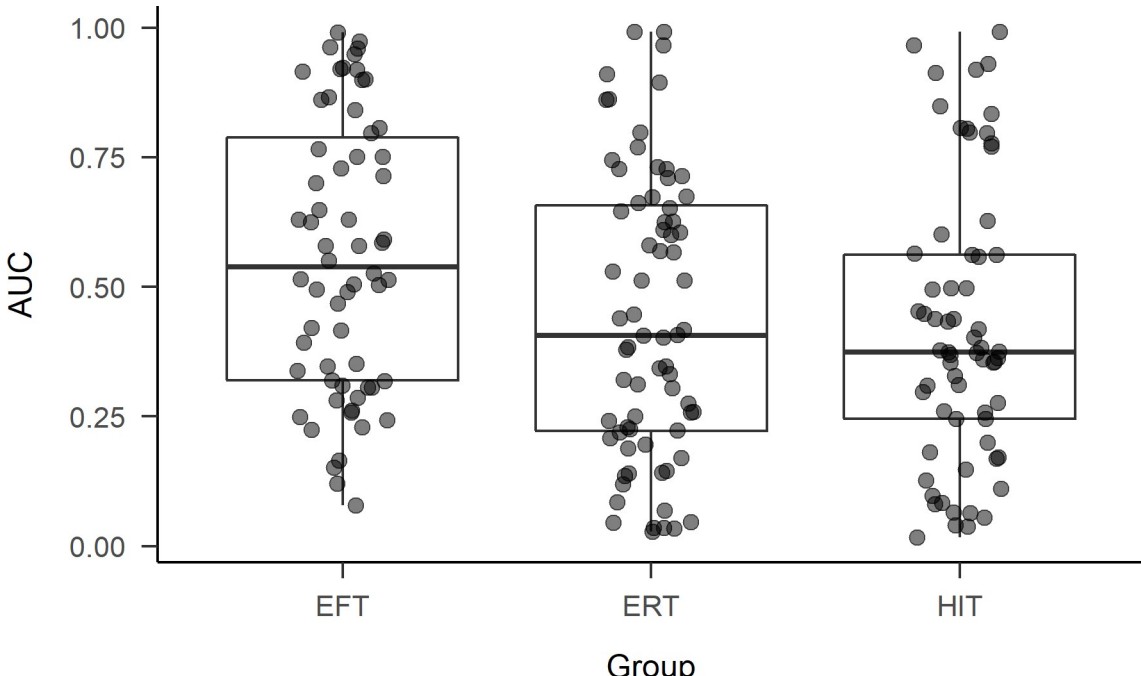

**Fig 1. Area under the curve (AUC) by assigned thought task (Group).** Boxplots of AUCs for participants assigned to the EFT, ERT, and HIT groups. Values closer to 1.0 indicate less steep discounting (or less impulsive choice) overall across large reward delays.

models, both pre- and post-diagnostics, are shown in Table 2. There was an overall significant effect of Group on AUC ($\chi^2[2]$ = 9.77, $p$ = .008), with those in the EFT group ($Mdn$ = .54) having significantly greater AUCs than those in the ERT ($Mdn$ = .41; $z$ = 2.61, $p$ = .014; $d$ = 0.44) and HIT groups ($Mdn$ = .37; $z$ = 2.89, $p$ = .012; $d$ = 0.52). Discounting did not significantly differ across those in the HIT and ERT groups ($z$ = -0.34, $p$ = .737; $d$ = -0.07). This pattern of results was the same in the post-diagnostics model (see Table 2) and all of the secondary models when controlling for positive and negative affect, vividness and frequency of thoughts, and differences in the duration it took to complete the thought task (see Table 3 for parameter estimates from secondary models with covariates).

While discounting did not differ across the HIT and ERT groups, the exploratory equivalence tests revealed that discounting across these groups was also not statistically equivalent.

**Table 2. Parameter estimates for the primary beta regressions predicting AUC.**

| Model/parameter | Beta | Std. error | Z | p |
|---|---|---|---|---|
| Model 1 (all observations) | | | | |
| Intercept (HIT) | -0.25 | 0.13 | -1.94 | .052 |
| ERT | 0.06 | 0.18 | 0.34 | .737 |
| EFT | 0.53 | 0.19 | 2.85 | .004 |
| Model 2 (post-diagnostics) | | | | |
| Intercept (HIT) | -0.36 | 0.13 | -2.86 | .004 |
| ERT | 0.06 | 0.17 | 0.34 | .737 |
| EFT | 0.66 | 0.18 | 3.65 | < .001 |

All parameter estimates represent the log-odds change in AUC.

**Table 3. Parameter estimates for the secondary beta regressions predicting AUC.**

| Model/parameter | Beta | Std. error | Z | p |
|---|---|---|---|---|
| Model 1 (Affect) | | | | |
| Intercept (HIT) | -0.62 | 0.33 | -1.85 | .064 |
| ERT | <0.01 | 0.18 | 0.01 | .992 |
| EFT | 0.49 | 0.19 | 2.54 | .011 |
| Positive Affect | 0.16 | 0.08 | 1.98 | .047 |
| Negative Affect | -0.13 | 0.11 | -1.19 | .234 |
| Model 2 (Thought Engagement) | | | | |
| Intercept (HIT) | -0.24 | 0.41 | -0.57 | .567 |
| ERT | 0.05 | 0.18 | 0.29 | .770 |
| EFT | 0.53 | 0.19 | 2.77 | .006 |
| Vividness of thought | -0.18 | 0.19 | -0.91 | .362 |
| Frequency of thought | 0.18 | 0.19 | 0.93 | .354 |
| Model 3 (Thought Task Duration) | | | | |
| Intercept (HIT) | 0.14 | 0.19 | -0.73 | .467 |
| ERT | 0.01 | 0.19 | 0.05 | .964 |
| EFT | 0.49 | 0.19 | 2.52 | .012 |
| Duration in thought task | <-0.01 | <0.01 | -0.86 | .391 |

All parameter estimates represent the log-odds change in AUC.

Using a threshold of +/- .10 as the range of equivalence, 94% of the posterior distribution of mean differences fell within this range. However, the 95% credible interval (most likely values in the mean difference distribution) fell slightly below this range (95% CI: -0.118–0.071), indicating that the true mean difference is slightly less than zero (estimated $\bar{x}_{diff}$ = -0.02) or, that the HIT procedure produced slightly steeper discounting relative to ERT. This result was nominally the same using a narrower credible interval (90% CI: -0.102–0.056), although the most credible mean differences nearly entirely fell within the range of equivalence.

## Discussion

The present research found that EFT reduced discounting relative to both a typical control (ERT) and a novel control more amenable to use in clinical contexts (HIT). Compared to ERT, the HIT procedure produced some affective and implementation differences. Namely, spending time thinking about recently acquired health information led to slightly less positive affect relative to EFT and ERT, although the aforementioned difference was nominal. While EFT was effective in reducing discounting regardless of control procedure, and differences between the ERT and HIT procedures were small (*Mdn* difference of obtained data = .04), results showed the two controls were not equivalent in their effect on discounting.

It is unknown from the present research what processes HIT engages; the information chosen for HIT was putatively novel, and casual review of participants' HIT cues supports this. Many individuals wrote commentary indicating that they either were unaware of the information presented or in some cases had thoughts to the contrary (e.g., individuals thinking alcohol helps or has no disruptive effect on sleep). To the extent that thinking about new information, and imagining the process of acquiring said information, could be considered episodic would make the HIT procedure comparable to EFT in terms of mechanisms engaged. To better understand the extent to which the HIT procedure engenders episodic processes versus those more commonly associated with factual information (e.g., semantic) necessitates direct

assessment and remains a topic for future investigation; initiating this translational research in non-clinical samples may be best, and later incorporation of clinical samples could provide conjoint means of testing theory (e.g., enrolling a clinical sample with poor episodic memory) and feasibility. Ultimately, research evaluating the design of control groups not only has the potential for supporting applied translation, but also for furthering our theoretical and mechanistic understanding of interventions.

Regardless of the processes engaged, the HIT procedure appears well-suited to the task of serving as a clinical control. First, the HIT procedure bears structural resemblance to EFT. Tests of psychotherapies that do not adequately control for structure of the therapy in control conditions reveal larger treatment effects, suggesting a sizeable portion of the resulting therapeutic effects may be due to nonspecific or atheoretical factors [36]. In this regard, the HIT procedure is a particularly rigorous control for the active ingredients of EFT in that its implementation was designed to be as close to EFT as possible (e.g., self-generation of tags and cues, discussion of thoughts and feelings within cues, etc.). While ERT in the lab shares structural similarity to EFT, its use in a clinical context—for an intervention that may span weeks, or months—is impractical if the temporal proximity of ERT events is to be maintained. Those assigned to an ERT condition in a clinical intervention would need to frequently regenerate ERT cues to maintain their temporal proximity. This frequency would then need to be matched by those in the EFT group, likely making the intervention burdensome.

Second, psychoeducational controls may have some modest therapeutic effect beyond a usual care, no-treatment or wait-list control [37]. As such, the use of a psychoeducational control is not uncommon when one desires to control for nonspecific factors of a treatment (see Chiesa and colleagues [38] for an example) so as to gauge the direct relevance of an intervention's theoretical ingredients. Utilizing a control procedure that captures some or all of the nonspecific factors of an intervention is important to determine the efficacy of a treatment; however, what specific aspects are important to control for and when is context-dependent and often a source of debate (see Baskin and colleagues [36] and Wampold, Minami, Tierney, Baskin, and Bhati [39]). While not directly tested herein, the HIT procedure is likely a better control for expectation and other nonspecific factors than ERT [19], which does not engender a similar expectancy of change/effect.

This early-stage translational study presents an initial demonstration of the feasibility of the HIT procedure in a non-clinical sample, but several limitations will need to be addressed in further research. Researchers may need to make procedural modifications to better equate thought task durations, cue use and vividness of thought, and affective measures across EFT and HIT groups. That the inclusion of the aforementioned variables in regression models did not impact the effectiveness of EFT relative to control groups suggests differences in these variables may not pose substantive concern. However, it is unknown if they are relevant to other dependent variables (e.g., alcohol [10] and cigarette use [7, 11]) or within clinical populations that are of candidate interest for EFT-based interventions. Online platforms such as AMT are incredibly helpful in the early stages of translational research, and utilizing them to test modifications to equate the characteristics of the content yielded by the HIT or related procedures is reasonable in this stage of translation.

In moving further down the translational continuum, the HIT control will need to be evaluated directly in clinical populations of interest (and its content modified for its particular application). AMT workers differ from the general United States population: they are generally younger, have lower yearly income but higher education, and are less likely to attend religious services among other characteristics [40]. Relatedly, consideration must be given to participants' naïveté to study procedures and tasks; while this did not appear to interfere with the

dependent measures herein, methods of gauging experimental histories should be devised when using crowdsourcing methods and these data explored in relation to study outcomes.

To fully understand the adequacy of the HIT procedure to control for nonspecific treatment effects on discounting, a more "neutral" control group may need to be employed. In the present research, the effects of EFT were evaluated relative to both HIT and the "gold-standard" laboratory control (ERT). HIT and ERT did not result in equivalent (or different) degrees of discounting—if anything, HIT may have a slight nocebo effect (i.e., a decrement in treatment effect, revealed by nominally greater discounting and/or less variability in HIT vs. ERT). Determining what, if any nonspecific effects HIT has on discounting requires comparison relative to an activity that controls for the passage of time without introducing theoretically relevant processes (i.e., a laboratory analogue to a wait-list control). The development of this type of control will allow for more theory-driven testing (and boundaries) of proper control groups. ERT represents an ideal laboratory control in that it controls for episodic thinking, the temporal perspective of thought, and the effects of ERT itself on discounting are minimal (ERT has an effect of approximately $d$ = .02 relative to a do-nothing control; Sze et al. [9]). This latter characteristic of ERT thus suggests that (1) ERT does not induce non-specific effects comparable to some clinical controls (in further support of Rung and Madden [19]), and (2) testing equivalency of novel clinical controls against ERT may not represent the best test of the utility of HIT for clinical research.

The lack of a non-specific effect on discounting resulting from HIT does not preclude its adequacy in a clinical context. While part of the empirical excitement of EFT is its robust ability to modify a decision-making process [3] the ultimate outcomes of interest are the health behaviors thought to be a function of steep discounting. Thus, the better gauge for the presence and control of non-specific effects may be through examination of HIT effects on associated health behaviors. Regardless, so long as the HIT procedure reasonably controls for time in treatment, engages some comparable cognitive processes to EFT, and provides expectancy for treatment success, HIT may be a useful control condition for EFT.

As research progresses on bringing the promise of EFT interventions into field and clinical settings, more research focused on factors critical in the demonstration of clinical utility and efficacy must be undertaken. At this time, the validity [19, 20] and robustness of EFT to a variety of different factors has been demonstrated (e.g., populations, health behaviors, task types, settings; see Rung and Madden [20, 41] for discussions), with the majority of these studies focused on concerns specific to basic research. Continued modifications in design to address clinical concerns will bolster the tools with which researchers may combat the problems associated with steep discounting; and set precedent for the formal development of a technology to do so.

## Supporting information

**S1 File.**
(DOCX)

## Acknowledgments

The authors would like to thank Niki Pradhan and Deana Chan for their assistance with data collection, screening, and processing.

## Author Contributions

**Conceptualization:** Jillian M. Rung, Leonard H. Epstein.

**Formal analysis:** Jillian M. Rung.

**Investigation:** Jillian M. Rung.

**Methodology:** Jillian M. Rung.

**Resources:** Leonard H. Epstein.

**Writing – original draft:** Jillian M. Rung, Leonard H. Epstein.

**Writing – review & editing:** Jillian M. Rung, Leonard H. Epstein.

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
