## [Decision Letter · Decision Letter 0]

25 Feb 2020

PONE-D-19-31639

Translating Episodic Future Thinking Manipulations for Clinical Use: Development of a Clinical Control

PLOS ONE

Dear Dr. Rung,

Thank you for submitting your manuscript to PLOS ONE. After careful consideration, we feel that it has merit but does not fully meet PLOS ONE’s publication criteria as it currently stands. Therefore, we invite you to submit a revised version of the manuscript that addresses the points raised during the review process.

We would appreciate receiving your revised manuscript by Apr 10 2020 11:59PM. To enhance the reproducibility of your results, we recommend that if applicable you deposit your laboratory protocols in protocols.io, where a protocol can be assigned its own identifier (DOI) such that it can be cited independently in the future. For instructions see: http://journals.plos.org/plosone/s/submission-guidelines#loc-laboratory-protocols

We look forward to receiving your revised manuscript.

Kind regards,

Gian Mauro Manzoni, Ph.D., Psy.D.

Academic Editor

PLOS ONE

Journal Requirements:

2. In keeping with usual publication standards, please remove trademark symbols from the manuscript.

**Comments to the Author**

1. Is the manuscript technically sound, and do the data support the conclusions?

Reviewer #1: Yes

Reviewer #2: Partly

2. Has the statistical analysis been performed appropriately and rigorously?

Reviewer #1: Yes

Reviewer #2: Yes

3. Have the authors made all data underlying the findings in their manuscript fully available?

Reviewer #1: Yes

Reviewer #2: Yes

4. Is the manuscript presented in an intelligible fashion and written in standard English?

Reviewer #1: Yes

Reviewer #2: Yes

5. Review Comments to the Author

Reviewer #1: This is an interesting study in which the investigators compare an EFT condition to two different control conditions to establish the utility of baseline comparisons for a well-established delay discounting effect. The method, analytic choices, and results were well motivated, easy to follow, and reproducible. The findings and discussion of them appear well-grounded. The conclusions do not go beyond the scope of the data. There were a few points that could be strengthened to elevate the rigor of the manuscript.

Introduction:

-In summarizing control conditions against EFT, it may be worth citing work from Benoit and colleagues who have used an ‘estimate’ control condition against EFT. Citing this work would situate the current manuscript in the appropriate literature context.

Method:

-While not a flaw of the manuscript, it struck me that the health control condition against EFT involves very different cognitive processes. For example, the health control condition by and large involves reading novel (not necessarily positive) information whereas EFT involves generating positive information. Has any work investigated whether generation of EFT is necessary? Would just reading specific future events from other participants have the same downstream consequences as generating specific future events? This could be cited as an avenue for future research to identify mechanisms of change involved in EFT vs. various controls.

Results:

-The sample had some caveats. For example, 30% of the sample had done EFT/ERT before, and 50% of the sample had done discounting tasks before. The investigators note that these data points are not flaws, and that previous discounting behavior doesn’t impact current discounting behavior. I thought motivation could be stronger here by citing empirical pieces that support the investigators’ claims. I also think the discussion should note that the sample was not completely naïve to study design or tasks.

Discussion:

-The abstract cites ‘non-specific’ treatment factors as related to the health control. What are the non-specific factors? I thought this line of thinking could’ve played a more prominent role in the discussion section.

Reviewer #2: The authors are to be commended for this study of HIT given that control conditions are not often directly studied but have important implications for designing clinical trials effectively. At the same time, I wanted a more compelling rationale for the need and benefits of this specific, novel HIT control, and I had some concerns about generalizability of the findings beyond the mTurk sample. I have detailed my primary comments below.

I think more background is needed on why the ERT procedure is problematic. For instance, it was not clear to me why it was so burdensome to regenerate events and cues repeatedly. More generally, a clearer description of the EFT and ERT procedures earlier would help better situate the need for a novel control condition, as would a little more information on how the EFT is used as an intervention and its theoretical mechanisms of change.

Analogously, more information was needed about what EFT task/intervention features the HIT controls for more effectively than the ERT controls for because this was not obvious to me.

I appreciated that the authors conducted a power analysis to determine sample size, but was not totally clear what was meant re. the “magnitude of the EFT effect” – is this referring to a group difference effect on some parameters of the EFT or a result when the EFT is used as an intervention…? My assumption is this referred to the difference in AUC on the discounting task when it had been paired with EFT relative to ERT but I was not sure.

This is minor but it was not until the Results on p. 13 that I knew what the sample size actually was – typically this is reported in the Abstract and in the Participants section.

It is important to discuss the limitations and implications of using an mTurk sample that is not clinical in any way. It seems hard to evaluate the clinical utility of a paradigm -a stated primary aim of the current study- without using an analogue or clinical sample.

More broadly, before claims about the clinical utility of the HIT control are warranted, the paradigm requires replication in another sample and tests of inter-rater and test-retest reliability and sensitivity to change, etc.

Taken together, I appreciate that the authors are directly evaluating different control conditions and think this is a valuable line of work, but I have concerns about what can reasonably be concluded from the current study as a stand-alone study.

---

## [Author Response · Author response to Decision Letter 0]

27 Mar 2020

Please see the attached "responses to reviewers" document.

---

## [Decision Letter · Decision Letter 1]

23 Jul 2020

PONE-D-19-31639R1

Translating Episodic Future Thinking Manipulations for Clinical Use: Development of a Clinical Control

PLOS ONE

Dear Dr. Rung,

Thank you for submitting your manuscript to PLOS ONE. After careful consideration, we feel that it has merit but does not fully meet PLOS ONE’s publication criteria as it currently stands. Therefore, we invite you to submit a revised version of the manuscript that addresses the points raised during the review process.

Two minor points on the manuscript:

The abbreviation HIT is used for two different things; perhaps just refer to human intelligence tasks throughout without abbreviating (or use a phrase like “AMT task” or similar) to avoid confusion.

I think it should be “*bears* structural resemblance” (line 424)

We look forward to receiving your revised manuscript.

Kind regards,

Andrew Allen

Academic Editor

PLOS ONE

Additional Editor Comments (if provided):

The authors have addressed the peer review comments in a structured manner, and I believe the article is almost ready for publication.

Two minor points on the manuscript:

The abbreviation HIT is used for two different things; perhaps just refer to human intelligence tasks throughout without abbreviating (or use a phrase like “AMT task” or similar) to avoid confusion.

I think it should be “bears structural resemblance” (line 424)

Reviewers' comments:

Reviewer's Responses to Questions

**Comments to the Author**

1. If the authors have adequately addressed your comments raised in a previous round of review and you feel that this manuscript is now acceptable for publication, you may indicate that here to bypass the “Comments to the Author” section, enter your conflict of interest statement in the “Confidential to Editor” section, and submit your "Accept" recommendation.

Reviewer #1: All comments have been addressed

2. Is the manuscript technically sound, and do the data support the conclusions?

Reviewer #1: (No Response)

3. Has the statistical analysis been performed appropriately and rigorously? 

Reviewer #1: (No Response)

4. Have the authors made all data underlying the findings in their manuscript fully available?

Reviewer #1: (No Response)

5. Is the manuscript presented in an intelligible fashion and written in standard English?

Reviewer #1: (No Response)

6. Review Comments to the Author

Reviewer #1: (No Response)

7. PLOS authors have the option to publish the peer review history of their article (what does this mean?). If published, this will include your full peer review and any attached files.

Reviewer #1: No

---

## [Author Response · Author response to Decision Letter 1]

24 Jul 2020

Responses included in attached word document.

---

## [Editor Report · Decision Letter 2]

28 Jul 2020

Translating episodic future thinking manipulations for clinical use: Development of a clinical control

PONE-D-19-31639R2

Dear Dr. Rung,

We’re pleased to inform you that your manuscript has been judged scientifically suitable for publication and will be formally accepted for publication once it meets all outstanding technical requirements.

Kind regards,

Andrew Allen

Academic Editor

PLOS ONE
---

## [Editor Report · Acceptance letter]

6 Aug 2020

PONE-D-19-31639R2 

 Translating episodic future thinking manipulations for clinical use: Development of a clinical control 

Dear Dr. Rung:

I'm pleased to inform you that your manuscript has been deemed suitable for publication in PLOS ONE. Congratulations! Your manuscript is now with our production department. 

Kind regards, 

on behalf of

Dr. Andrew Allen 

Academic Editor

PLOS ONE